# An Extensive Assessment of Network Embedding in PPI Network Alignment

**DOI:** 10.3390/e24050730

**Published:** 2022-05-20

**Authors:** Marianna Milano, Chiara Zucco, Marzia Settino, Mario Cannataro

**Affiliations:** Department of Medical and Surgical Sciences, Data Analytics Research Center, University Magna Græcia, 88100 Catanzaro, Italy; chiara.zucco@unicz.it (C.Z.); marzia.settino@unicz.it (M.S.); cannataro@unicz.it (M.C.)

**Keywords:** network embedding, network alignment, PPI

## Abstract

Network alignment is a fundamental task in network analysis. In the biological field, where the protein–protein interaction (PPI) is represented as a graph, network alignment allowed the discovery of underlying biological knowledge such as conserved evolutionary pathways and functionally conserved proteins throughout different species. A recent trend in network science concerns network embedding, i.e., the modelling of nodes in a network as a low-dimensional feature vector. In this survey, we present an overview of current PPI network embedding alignment methods, a comparison among them, and a comparison to classical PPI network alignment algorithms. The results of this comparison highlight that: (i) only five network embeddings for network alignment algorithms have been applied in the biological context, whereas the literature presents several classical network alignment algorithms; (ii) there is a need for developing an evaluation framework that may enable a unified comparison between different algorithms; (iii) the majority of the proposed algorithms perform network embedding through matrix factorization-based techniques; (iv) three out of five algorithms leverage external biological resources, while the remaining two are designed for domain agnostic network alignment and tested on PPI networks; (v) two algorithms out of three are stated to perform multi-network alignment, while the remaining perform pairwise network alignment.

## 1. Introduction

Different complex systems can be modelled as networks [1]. For instance, interactions between proteins inside an organism are modelled as protein–protein interaction (PPI) networks. A comparative analysis of PPI networks across different species is known as network alignment (NA) [2]. NA searches for the best fit among the nodes of compared networks with the aim to find network regions of high similarity [3]. The regions with high similarity correspond to interaction patterns that are conserved during evolution. This enables us to transfer biological knowledge from well-studied species, i.e., yeast, to species whose studies are still ongoing, such as humans. In fact, by aligning the PPI network of yeast with the PPI network of the human, it is possible to infer the function of human proteins based on the function of their aligned counterparts in the yeast network. Consequently, this leads to new discoveries in system biology. Diverse network alignment approaches have been used in the biological domain [4]. According to the type of input, the network alignment approaches can be categorized as *pairwise or multiple alignment*, whereas, according to the scope of node mapping desired, the network alignment methods can be classified as *global network alignment (GNA) or Llocal network alignment (LNA)*. GNA algorithms search for the best alignment of entire networks, whereas LNA algorithms produce the aligned pairs of small sub-networks. As introduced before, the network alignment methods can be classified based on the number of networks on which they build the alignment, i.e., *pairwise network alignment (PNA) and multiple network alignment (MNA)*. PNA algorithms build the alignment of two networks and enable us to detect the same regions among the analyzed networks. MNA algorithms align three or more networks with each other and build an alignment consisting of aligned node clusters.

Despite the existence of different approaches, there are some issues related to network representation that still need to be solved. The network representation enables us to highlight important interactions between entities compared to the analysis of each one.

However, the traditional network modelling still presents critical problems in large-scale network analysis. A first issue is related to high computational complexity. In fact, in large networks, the high number of nodes and edges affects most of the analysis algorithms in their iterative or combinatorial steps. Thus, these methods result in high computational complexity, and they cannot be applied to large-scale networks. A second issue is related to the lack of parallel and distributed algorithms for the analysis of traditional network representation. Although these methods are suitable to analyze large-scale data, the parallelization of classical network alignment causes high costs among servers and degrades the speed-up ratio, due to the network topological characteristics. The third issue is related to the inapplicability of machine learning methods to analyze large-scale networks. The reason is that these methods require the representation of data as independent vectors in a vector space. Instead, in the classical network representation, the nodes are dependent on each other according to such degree. Thus, in a large network, the high dimensionality derived from many nodes causes the analysis to be problematic.

On the other hand, it is widely recognized that a fundamental role in predictive analytics is played by data representation [5]. One of the reasons linked to the popularity and the effectiveness of deep learning algorithms is that while solving a specific task, deep learning algorithms jointly aim at learning a latent representation of the data, without strongly relying on hand-crafted feature engineering, typical of classic machine learning and data mining processes.

Inspired by the success achieved in different domains, and especially in Natural Language Processing (NLP), several methods have been proposed to learn a meaningful and dense mapping of graphs in lower-dimensional, latent vector spaces. Here, the representation is deemed to be meaningful if it represents each node in a network as a low-dimensional feature vector such that the network structure is preserved, while dense implies that the vector space is sought to have a smaller dimension than the number of nodes in the starting graph. The learned mapping is also known as network embedding or network representation.

Although several network embedding methods have shown state-of-the-art performance in different tasks such as, for instance, link prediction, node clustering, and node classification, some works have questioned whether representation learning should lead to some improvements in the context of inter-network problems, of which network alignment is one of the most representative [6,7].

The main focus of the present paper is to review existing representational learning approaches for the problem of aligning pairwise or multiple PPI networks. To this end, we will start by presenting an overview of classical PPI network alignment techniques, providing a traditional classification and describing the common framework used to evaluate classical network alignment techniques. Then, we will review five approaches that leverage different network embedding techniques for the network alignment (NE-NA) of PPI networks. All the considered NE-NA algorithms present two main blocks, the first of which generates a node embedding of the considered networks in the same vector space, while the second one exploits node representations to perform the alignment. On the basis of the embedding, the considered algorithms may be classified into matrix factorization and diffusion methods, while the alignment block may rely on geometric or optimization approaches. Another main issue strictly related to the NE-NA problem may be related to the lack of a common framework to compare performance. For this reason, the present review will focus on the evaluation metrics and compare them with the alignment evaluation metrics typical of a classical setting. The paper is organized as follows: in Section 2, an overview of the existing literature reviews on network representation is presented, and the main differences among them and the present review are highlighted; in Section 3, the classical framework of network alignment is discussed, while in Section 4, the problem of representation learning is tackled, with a major focus on the overview of five different algorithms that have addressed the problem of exploiting network embedding (NE) for network alignment (NA) in the context of PPI networks. In Section 5, the main differences among classical NA and NE-NA approaches are discussed, and a qualitative comparative review is carried out for the five NE-NA algorithms considered. Finally, Section 6 concludes the paper.

## 2. Related Surveys and Differences

The literature contains different works that explore the network embedding problem. To date, not many works provide a comprehensive survey of representation learning and most of them concern non-biomedical tasks such as social networks and citation networks.

In this section, we explore the works that introduced only the representation learning and the network representation learning, and then we examine the works that proposed a taxonomy for machine learning on graphs and the analytical task. The fist work on network representation learning (NRL) was presented by Luis G. Moyano [8]. The author introduced some key notions of representation learning and an overview of the most widespread methods for NRL. Finally, Moyano described the application of NRL in various fields. Subsequently, Hamilton et al. [9] provided a review of advancements in representation learning approaches for machine learning on graphs by introducing matrix factorization-based methods, random-walk based algorithms, and the graph neural network-based approach for embedding subgraphs. Furthermore, for the first time, they presented the main applications of node embeddings, i.e., visualization, clustering, node classification, and link prediction. With their work, Zhang et al. [10] introduced two important innovations. At first, they provided a review of the state-of-the-art network representation learning techniques. On the basis of this, they proposed a new taxonomy with the goal to summarize existing techniques and to illustrate the advantages and disadvantages of different algorithms. Then, Zhang et al. provided a summary of protocols used for validating the effectiveness of NRL techniques and of commonly used benchmark datasets that also include biological networks. Finally, they presented analytic tasks, such as node classification, link prediction, clustering, visualization, and recommendation. Later, three works introduced important advances. Chen et al. [11] provided an overview of network embedding by focusing on supervised and unsupervised learning for homogeneous and heterogeneous networks. Furthermore, the authors presented a detailed review of several representative applications of network embeddings. Goyal et al. [12] proposed a taxonomy of graph embedding techniques covering three main categories of approaches: factorization, random walks, and deep learning. Furthermore, they compared the performance of the surveyed methods on both synthetic and real datasets, as well as providing a broad classification of graph embedding applications: network compression, visualization, clustering, link prediction, and node classification. Moreover, the authors made publicly available an open-source library for implementing all the presented methods, including their evaluation metrics. Cui et al. [13] discussed the relationship and the critical differences between classical graph embedding and network embedding. Furthermore, they reviewed network embedding methods by focusing on the approaches that preserve the network structure and properties, as well as providing an evaluation of these methods applied on different network datasets. Moreover, they introduced the concept of advanced information preserving network embedding, which consists of two parts: one is to learn the network representation while preserving its structure; the other is to align the resulting embedded network to a target-task network using a network embedding algorithm (e.g., to predict the anchor links across social networks).

A benchmark of some of the most popular real-world networks used in the network embedding literature was provided. Their related datasets are divided into four groups (i.e., social, citation, language, biological networks), according to the nature of the networks. Evaluation tasks, commonly used in network embedding, are summarized (i.e., node classification, link prediction, clustering, and visualization).

Furthermore, they proposed two taxonomies of graph embedding based on problem settings (i.e., embedding input and embedding output) and categorized the literature based on embedding techniques. The authors provided a categorization of the applications that graph embedding enables and, for each category, they presented application scenarios as a reference.

Nelson et al. [7] provided a a discussion with a biological slant by reviewing different approaches for graph embedding, including spectral-based, diffusion-based, and deep-learning-based methods, as well as presenting their application in a variety of biological applications ranging from protein network alignment and protein function prediction to community detection and network denoising.

Yue et al. [14] investigated the effectiveness and potential of advanced graph embedding methods and their applications in three important biomedical link prediction tasks, namely drug–disease association (DDA) prediction, drug–drug interaction (DDI) prediction, and protein–protein interaction (PPI) prediction, and two node classification tasks, namely medical term semantic-type classification and protein function prediction.

The authors included seven benchmarks on datasets for all the addressed prediction tasks and used them to evaluate eleven representative graph embedding methods. Moreover, they provided some guidelines for the choice of embedding method. Furthermore, they presented an easy-to-use Python package developed for facilitating the study of various graph embedding methods on biomedical tasks.

As can be deduced from the reported literature, a limited number of surveys have shed light on the graph embedding approach according to a biological perspective.

Nelson et al. and Yue et al. are the only studies that provide graph embedding evaluations and analyses on biomedical networks. However, both have some limitations and gaps. The first provides a review of graph embedding methods by comparing their application to biological issues.

However, some of the selected methods are highly dependent on the considered biomedical task and may not be generalized to other tasks. The latter conducts a comparison of the most representative graph embedding methods on meaningful biomedical link prediction tasks, including protein–protein interaction (PPI) prediction. However, because of the broad scope of the addressed issue, a in-depth examination entirely focused on PPI network embedding may be helpful for researchers and practitioners.

Furthermore, none of the considered surveys address the issue of network embedding in order to learn a network representation that best fits the goal of aligning different biological networks. This work aims to fill this gap providing a comprehensive survey on network embedding methods best-suited to the comparative analysis of PPI networks through network alignment techniques.

## 3. Network Alignment

Network alignment (NA) is a computational technique widely used for the comparative analysis of PPI networks between species, in order to discover the similarities between the molecular systems of different organisms. Network alignment is a common problem that seeks to determine how to best fit one network into another network. This ’fit’ can be quantified by assigning the amount of conserved, and hence aligned, edges. Otherwise, a measure of node conservation (NC) quantifies the similarity between pairs of nodes from different networks. In other words, intuitively, a good alignment should both map similar nodes to each other and preserve many edges. Thus, the problem of graph alignment consists of the mapping between two or more graphs to maximize an associated cost function (also known as quality of alignment) that represents the similarity among nodes or edges. Conventionally, given two graphs, $G_{1}=\left\{V_{1},E_{1}\right\}$ and $G_{2}=\left\{V_{2},E_{2}\right\}$, the graph alignment searches an alignment function $f:V_{1}\to V_{2}$, where $V_{1,2}$ are a set of nodes, that maximizes the quality of alignment. The size of this search space is large, as it consists of all possible mappings between the nodes of the compared networks. The computational intractability of the above problem, which arises from the NP-completeness [15] of the underlying subgraph, or the *subgraph isomorphism issue*, requires the development of heuristics (approximate approaches) to solve the problem. The **alignment quality** is a function of cost that measures the level of similarity of the analyzed network, and it is defined as follows: $Q(G_{1},G_{2},f)$. Q conveys the correspondence among the input networks on a precise alignment *f*. Thus, the Q definition highly affects the mapping approach [16]. In the literature, there exist different network alignment algorithms that investigate approximate solutions. The network alignment algorithms can be categorized according to the number of input networks, i.e., pairwise and multiple alignment, and the goal of node matching, i.e., local or global alignment.

### 3.1. Global or Local Alignment

In general, network alignment can be classified as global alignment and local alignment.

Global network alignment (GNA) algorithms search for the best mapping that covers all nodes of the input networks by generating a one-to-one node mapping. This strategy considers only the topology of input graphs, leaving out the similarity among small regions (representing conserved motifs). In general, GNAs exploit a two-step schema to build the alignment. At first, the algorithms adopt a cost function, which maximizes the node likeness, also known as node preservation, or the quantity of preserved edges, i.e., edge preservation, to estimate the similarity among pairs of nodes. Then, they apply an alignment method to find a high-scoring alignment based on the total similarity over the all aligned nodes among all the possible alignments. For example, IsoRank [17] maximizes an alignment quality measure that balances topological and node similarity using a weight factor α. In addition, there exist different algorithms, such as GRAAL [18] and the GRAAL family (H-GRAAL [19], MI-GRAAL [20], C-GRAAL [21], L-GRAAL [22]), that use a special node similarity measure called the graphlet degree vector [23] to build a global alignment. The main characteristic of the graphlet degree vector is the generalization of the node degree, by counting the degree values for all possible connected induced subgraphs up to a certain node number. GHOST [24] uses a novel spectral signature based on the local neighborhood’s topology to measure the topological similarity between subnetworks. The idea behind GHOST consists of building an alignment by combining the novel spectral signature with a seed-and-extend strategy. WAVE [25] uses a *seed-and-extend* alignment strategy to optimize both node and edge conservation while constructing an alignment. MAGNA [16] applies a genetic algorithm to build an improved alignment. MAGNA considers a set of alignments and it chooses the best among them. MAGNA++ [26] is the MAGNA extension and it aims to maximize any edge and node conservation measures. SANA (Simulated Annealing Network Aligner) [27] uses Simulated Annealing to build a final alignment, starting with two networks, and an input alignment randomly built or by applying different aligners. More recently, Malod-Dognin et al. [28] presented UAlign, which associates different alignments built by a global network aligner. The aim is to overcome the limitations of the global network aligners present in the coverage of built alignments. IGLOO [29] is able to achieve high functional and topological quality using a seed-and-extend strategy on the basis of the integration of the GNA and LNA strategies. Local network alignment algorithms (LNAs) search multiple small subnetworks with high similarity among input networks by producing a many-to-many node mapping. These subnetworks are conserved patterns of interaction that can correspond to preserved or activity patterns. LNAs employ a two-step schema to build the alignment. At first, they take as input a set of *seed nodes* chosen by biological information; then, the algorithms merge the inputs in a complementary structure, also called an alignment graph. Finally, LNAs mine the graph to extract interesting modules. NetworkBLAST [30] researches highly connected node groups corresponding to groups of proteins with the same function. NetAligner [31] presents a strategy to identify evolutionarily conserved interactions, on the basis of the consideration that interacting proteins evolve at rates significantly closer than expected by chance. AlignNemo [32] ensures the detection of subgraphs of proteins with similar biological functions according to the topology. AlignNemo can build the alignment of sparse PPI networks because it analyzes the topology of adjacent nodes of interacting proteins. AlignMCL [33] is the evolution of the previous AlignNemo. AlignMCL constructs the local alignment by integrating all the input data in the *alignment graph*, which is subsequently clustered by using the MCL algorithm [34] to mine the conserved subnetworks. GLAlign (Global Local Aligner) is a novel local network alignment methodology [2] that applies a global network algorithm to generate a list of seed nodes on the basis of topological information. Then, GLAlign integrates this topology information with biological information (i.e., homology relationships) by using a linear combination schema. At the end, GLAlign takes as input the generated global result for a local network aligner.

### 3.2. Pairwise or Multiple Alignment

A different way to formulate the network alignment problem considers the number of networks that need to be aligned: *pairwise or multiple alignment*. The pairwise network alignment (PNA) takes as input two networks and it identifies a subnetwork with high similarity among the input networks. The multiple network alignment (MNA) builds the alignment among three or more networks and it detects aligned patterns of nodes. PNA and MNA can be classified into the global approach by exploiting a many-to-many node mapping and the local approach by exploiting a one-to-one node mapping.

The PNA applies a many-to-many node mapping among the input networks, with the goal to find similar subgraphs. Otherwise, PNA exploits one-to-one node mapping to search the best match by considering the entire input networks. The MNA applies a many-to-many approach to find an alignment as a cluster that contains different nodes from one compared network. Otherwise, MNA exploits one-to-one node mapping to build an alignment as a cluster that contains only one node for each compared network. Though PNA and MNA have been applied on protein interaction networks (PINs) to build the alignment of [35], it has been shown that the alignment constructed with MNA is able to offer more biological knowledge since its approach is able to detect function information common to different species.

There exist different proposed multiple network alignment algorithms in the literature. MultiMAGNA++ [36] is a global MNA algorithm based on one-to-one node mapping. MultiMAGNA++ uses a genetic algorithm and introduces a new cost function to derive from the parent nodes a new child that allows the construction of subsequent multiple alignments. GEDEVO-M [37] is a global one-to-one MNA algorithm based on an evolutionary algorithm that uses the Graph Edit Distance (GED) as an optimization model for finding the best alignments. LocalAli [38] is a local many-to-many algorithm that uses an evolutionary model to derive an evolutionary tree of network nodes. Then, LocalAli builds the local alignment as conserved modules which descend from a common ancestral module. IsoRankN [17] is a global many-to-many algorithm that builds a multiple network alignment by using a spectral partitioning method to find dense and clique clusters. SMETANA [39] is a global many-to-many aligner that computes the node similarities using a probabilistic model and then applies a greedy approach to build a multiple alignment. FUSE [40] is a global one-to-one MNA algorithm that defines node similarities between all pairs of networks by applying a non-negative matrix trifactorization. After this, FUSE applies an approximate maximum weight k-partite matching algorithm to build an alignment between the multiple networks. NetCoffee [41] is a global many-to-many aligner that builds a weighted bipartite graph for every pair of networks and then it applies a simulated annealing approach to construct a multiple alignment. BEAMS [42] is a global many-to-many aligner that constructs a graph of node similarities considering protein sequence scores and detects from the graph a set of disjoint cliques that maximizes an alignment quality measure.

Figure 1 presents different examples of alignment types. Table 1 summarizes the different network alignment algorithms.

### 3.3. Quality Evaluation of Network Alignments

The evaluation of pairwise network alignment algorithms can be performed by considering both topological and biological aspects; see [43] for a complete discussion.

The topological quality is related to alignment algorithm capability as the reconstruction of the true node mapping and the conservation of as many edges as possible.

A widely used measure is *node correctness (NC)* [18], which is defined as the fraction of nodes of one network that correctly overlap with the nodes of a different network concerning the true node mapping. The NC measure is applied only in global network alignments. The reason is related to the approach on which the local alignment algorithm is based, i.e., the mapping of a node from a network with different nodes of the other network [4]. Subsequently, Meng et al. designed new measures for both global and local alignments: *P-NC*, *R-NC*, and *F-NC*. Let us consider an alignment *f* that produces a set of node pairs composed of $N_{al}$ nodes, while the true node mapping is composed of $M_{tr}$ nodes. *P-NC* is defined as $\frac{M_{tr}\cap N_{al}}{M_{tr}}$. *R-NC* is defined as $\frac{M_{tr}\cap N_{al}}{N_{al}}$. *F-NC* is the harmonic mean of the two previous measures.

Similarly, three popular measures to compute the edge correctness have been proposed: *edge correctness* (EC) [18], *induced conserved structure* (ICS) [24], and *symmetric substructure score* (S3) [16]. However, S3 can be used to measure the quality of global network alignment algorithms. For this reason, three measures have been designed [4], including *generalized S3* (GS3) and *high node coverage* S3 (NCV-S3).

The evaluation of alignment biological quality is performed by applying two measures: *Gene Ontology (GO) correctness*, which identifies the number of aligned protein pairs that are annotated with the same GO terms, and *Precision (P-PF), Recall (R-PF), and F-score (F-PF)*, which measure the accuracy of known protein function prediction. In particular, P-PF is the fraction of the union of the set of predicted protein–GO term associations and the set of true protein–GO term associations out of the set of predicted protein–GO term associations. R-PF is the fraction of the union of the set of predicted protein–GO term associations and the set of true protein–GO term associations out of the set of true protein–GO term associations. F-PF is the harmonic mean of P-PF and R-PF.

Moreover, a topological and functional quality assessment is performed for multiple alignment. The adopted measures to evaluate the topological alignment quality are: *adjusted node correctness (NCV-MNC), adjusted cluster interaction quality (NCV-CIQ), and largest common connected subgraph (LCCS)*.

NCV-MNC is the geometric mean of the node coverage (NCV), defined as nodes forming an alignment cluster with respect to all nodes in the networks, and cluster consistency (MNC) defined as one minus the mean of normalized entropy (NE) of all clusters in the alignment.

NCV-CIQ is the geometric mean of the node coverage (NCV) and CIQ, which measures edge conservation, and it is the generalization of the established S3.

LCCS is the geometric mean of the fraction of the number of nodes in the LCCS out of the maximum possible number of nodes in the LCCS and of the fraction of the number of edges in the LCCS out of the maximum possible number of edges in the LCCS.

To evaluate the functional alignment quality for MNA, three measures are defined: *Mean Normalized Entropy (MNE), GO correctness (GC), and accuracy of protein function prediction*.

MNE measures the internal cluster consistency and it is defined as the mean of the normalized entropy (NE) across all clusters in the alignment. In the multiple alignment, the NE takes into account the number of unique GO terms, the number of proteins annotated with a GO term, and the total number of protein–GO term annotations in the cluster.

GC is the fraction of protein pairs that share one or more GO terms.

Accuracy of protein function prediction is computed according to Precision (P-PF), Recall (R-PF), and F-score (F-PF), which are defined above. Table 1 summarizes the different network alignment algorithms and evaluation measures.

## 4. Representation Learning

One commonly used taxonomy classifies network embedding algorithms into three major groups: matrix factorization-based approaches, random walks or diffusion-based approaches, and deep learning-based approaches [7,9].

Matrix factorization-based approaches obtain the embedding through the factorization of a suitable matrix that encodes some graph property—for instance, an adjacency or a similarity matrix.

In analogy to word embedding algorithms, several network embedding algorithms that are generally categorized as random walks or diffusion-based approaches aim at sampling finite sequences of nodes from a network, and then perform the skip-gram algorithm to model the node context in a word2vec fashion [44].

The skip-gram model addresses the unsupervised problem of learning a text representation by simulating a supervised learning task to predict surrounding words given a word in input.

DeepWalk [45] is the first algorithm to be widely used as a benchmark in comparison with other graph learning approaches.DeepWalk randomly samples node sequences from each node by performing truncated random walks and then trains one hidden layer skip-gram algorithm to predict the node context. The representation is computed as the output of the hidden layer of the learned skip-gram model.

Node2vec [46] is a subtle modification of DeepWalk in which a second-order biased random walk is combined with Breadth-First Sampling (BFS) and Depth-First Sampling (DFS); the two extreme sampling strategies are combined to generate a neighborhood set of k nodes.

Deep learning-based methods exploit popular deep learning-related techniques, such as autoencoders, Convolutional Neural Networks (CNN), and Generative Adversarial Networks (GANs), to learn the latent representation of the network.

Although general-purpose network embedding methods may capture intra-network features, showing state-of-the-art performance in different tasks such as link prediction, node clustering, and node classification, the empirical analysis in [6] has shown that the node representation over different runs of the same algorithm may not correspond, or it may fail to preserve neighborhood consistency.

As a consequence, a specific network embedding approach needs to be designed to learn a robust and consistent representation, suitable for inter-network problems, of which network alignment is one of the most representative.

### Network Embedding for Network Alignment

Different network alignment approaches that leverage network representation have been proposed. In particular, CrossMNA [47] is an algorithm that performs two joint network embeddings to perform multi-network alignment. The novel idea behind CrossMNA is not to rely on topology consistency, i.e., the same node tends to maintain a similar connectivity structure across different networks.

DANA [48] and UAGA [49] propose two unsupervised network alignments that combine unsupervised graph embedding approaches with adversarial training.

However, none of the network embedding alignment strategies discussed so far have been tested in the context of biological networks, particularly PPI networks.

The present section discusses some network embedding algorithms for network alignment (NE-NA), which have been explicitly designed or tested on PPI networks. Table 2 summarizes the five considered works.

MUNK [50] is a semi-supervised node embedding approach that aims at jointly representing two (or more) PPI networks in a single vector space.

MUNK’s representation of PPI networks may be useful for cross-species inference, such as network alignment, multi-species synthetic lethality, and phenolog detection. However, for the purpose of the present review, only the NA problem will be taken into account.

Although, in the paper, the authors mention that MUNK can be used for MNA purposes, the algorithm is solely presented in a pairwise fashion and, to date, also the provided implementation only supports PNA.

MUNK receives as input a PPI source network and a target PPI network, a list of homologous protein pairs, also called landmarks, which guide the joint representation of the two networks, and a kernel function that captures the intra-network similarity of nodes. In the experiments, the authors use the regularized Laplacian [51] as a kernel function.

Initially, MUNK builds two kernel matrices, $D_{1}$ and $D_{2}$, having as matrix order n×n and m×m, respectively. Being symmetric and semi-definite positive, there exists a matrix $C\in R^{⋉\times }$ with d≤n such that $D_{1}=CC^{T}$.

The column vectors of *C* are assumed to be a representation of the nodes of the source network in the vector space $R^{d}$.

The vector representation $C_{2}$ of nodes belonging to the target network is reconstructed by constraining the landmarks to solve the following equation:$$C_{2}=C_{1L}^{*}D_{2L}$$
where $C_{1L}^{*}$ is the Moore–Penrose pseudoinverse of the landmark vector representation of *C*, while $D_{2L}$ represents the restriction of $D_{2}$ to the set of landmarks. The output of MUNK is the n×m MUNK similarity matrix $D_{12}$, computed as
$$D_{12}=CC_{2}^{T}$$
where the MUNK similarity score between node $v_{i}$ and node $v_{j}$ is expressed as $s_{ij}=c_{i}^{T}c_{j}$, where $c_{i}$ and $c_{j}$ are the learned representations for nodes $v_{i}$ and $v_{j}$, respectively.

To evaluate the representation performance in the context of NA, the author applied the Hungarian algorithm [52] to the MUNK similarity matrix to generate a matching set that maximizes the similarity scoring function.

The performance has been evaluated by considering the Jaccard index of the GO terms, also known as Gene Ontology consistency (GO Con), and the Area Under the Precision–Recall curve k-functional similarity (k-func AUPR).

Another comparison between Isorank and MUNK shows that while the former outperforms the latter in terms of EC, MUNK is 13 times faster in terms of runtime [7].

**Protein2Vec** [53] is an MNA algorithm that takes two or more PPI networks as input and outputs a set of matching protein complexes. In this context, Protein2Vec defines the similarity of two nodes as a convex combination of a topological and a biological similarity function. While the latter is based on a significance score of the Expect Values of pairwise proteins in BLAST, the former is the cosine similarity between the vector representation of protein nodes. Similarly to the previously described diffusion-based embeddings, Protein2Vec learns a network representation by generating context sequences for each node by biased random walks on an auxiliary structural similarity graph. At each layer, Protein2Vec assigns an iterative, normalized hierarchic variance weight, inspired by struc2vec [54] but further enhanced by adding a linear term that also considers the over-representation of triangle motifs. To perform the multiple network alignment, Protein2Vec applies a maximum weighted bipartite matching algorithm to optimize the number of pairs of vectors having the best similarity, and a simulated annealing algorithm.

Performance has been evaluated to assess the biological consistency of the alignment, by considering ME and MNE, as well as commonly used topological measures such as EC, ICS, and S3.

**GeoAlign** [55] is a many-to-many PNA algorithm designed to align PPI networks in two steps. The first step exploits the Structure Preserving Embedding (SPE) algorithm [56] to learn a low-dimensional vector representation that encodes the connectivity of the graphs in terms of the Euclidean distance among points in a vector space. To do so, given an adjacency matrix as input, the SPE algorithm constructs the embedding as the action of a positive semi-definite kernel matrix $K\in R^{N^{2}}$. The algorithm is constructed to be such that, given a connectivity algorithm G (e.g., KNN algorithm), the action of G on *K* reconstructs the adjacency matrix of the graph. In the second step, similarly to Protein2Vec [53], GeoAlign defines a matching score as a linear convex combination of a topological and a biological similarity score. In particular, the similarity scores among nodes as the flow value is calculated using the Network Simplex algorithm, while the topological similarity score is computed as the Earth Mover’s Distance between the embedding representation of the target network, and rigid transformations of the embedding representation of the source network. Finally, GeoAlign matches node *i* in the first network to node *j* in the second network if the similarity between the two nodes’ representation is greater than a threshold and outputs a list of detected node matching. Performance has been evaluated to assess the biological consistency of the alignment, i.e., the MNE, the specificity (Spe), which measures the ratio of correct clusters to annotated clusters, and the Conserved Orthogolous Interactions (COI), which measures the ratio of the total number of interactions between correct clusters to the number of aligned interaction. Moreover, ICS has been evaluated to assess the topological quality of the alignment.

**REGAL** [57] is a domain-agnostic NE-NA algorithm whose aligning strategy leverages a matrix factorization-based embedding to encode the topology-related information and, eventually, also some attribute information of nodes.

Given the scope of the present review, we will discuss the simplest case by assuming that no attribute information is provided. Then, the similarity between two nodes, nodes u and v, is defined as the exponential of the Euclidean distance between the degree vectors of node u and v, respectively, weighted by some parameter: $S\left(u,v\right)=e^{-\lambda \cdot \left|\right|d_{u}-d_{v}{\left|\right|}_{2}^{2}}.$

To compute *S* efficiently, REGAL proposes a low-rank approximation $S^{*}$, which takes advantage of a set of p landmarks. Let C be the similarity matrix between the joined set of vertices of the two networks $V_{s}⋂V_{t}$ and the set of landmarks, and let W be the similarity matrix between the landmark nodes. REGAL shows that matrix S is similar to the Moore–Penrose pesudoinverse of W. By performing a singular value decomposition of the pseudoinverse of W, the similarity matrix S can be computed as $S=YZ^{T}$. To compute the alignment for each node, the embedding of the target network is computed and stored in a *k*-d tree [58] to find the nearest (or sorting the k-nearest) node representation from the first network, with Euclidean distance. Two metrics have been considered to evaluate performance: Alignment Accuracy (or top-α accuracy) and runtime. Alignment accuracy (AA) has been defined as
$$AA=\frac{\#of\;correct\;aligments}{total\;\#\;of\;alignments},$$
while top-α accuracy (αAA) may be defined as
$$\alpha AA=\frac{\#of\;correct\;aligments\;in\;top-\alpha \;choices}{total\;\#\;of\;alignments}$$

**CONE-Align** [59] is a domain-agnostic, pairwise network alignment algorithm inspired by the unsupervised method used in Fasttext to align, in the same embedding space, two continuous word representations learned separately from two different languages [60].

In more detail, CONE-Align takes as input two adjacency matrices, and for each graph, it generates a separate node embedding by using NetMF [61], an embedding method which aims at approximating the DeepWalk matrix by explicitly factoring it through the SVD algorithm. Similarly to DeepWalk, NetMF learns an embedding that preserves node proximity. Then, CONE-Align aims at aligning the two graphs’ embedding subspaces $Y_{1},\;Y_{2}$ by solving the Wasserstain Procrustes problem [60]
$$min_{Q\in O^{d}}min_{P\in P^{n}}\left|\right|Y_{1}Q-PY_{2}\left|\right|.$$

Here, $O^{d}$ is the set of orthogonal matrices, while $P^{n}$ is the set of permutation matrices. Finally, the alignment is learned by matching each node in the representation of the first network to the nearest k-node representation of the second graph through a *k*-d tree search.

Performance has been tested on four different datasets, spanning from communication and social networks to PPI networks. Alignment Accuracy has been evaluated for each dataset.

## 5. Discussion

In this survey, we presented the network alignment problem by considering how the classical approaches and network embedding algorithms manage the alignment. By comparing these two different methods, at first, it possible to note that, in the literature, there exist several algorithms that are developed to handle the problem of the comparison of PPI networks via classical alignment. Otherwise, fewer network embedding alignment algorithms have been developed in recent years and very few of them are applied in the biological context. Furthermore, a major difference involves the type of output generated by two alignment strategies. While the classical network alignment algorithms generate many-to-many or one-to-one node mappings in the case of both PNA or MNA, these types of output are not obtained when network embedding alignment algorithms are applied. In fact, REGAL and GeoAlign produce as output a matching node list, MUNK generates a similarity score matrix, Protein2vec produces a similarity ranking, and CONE-ALIGN builds an alignment as a matching node list. Instead, both classical network alignment approaches and network embedding alignment strategies can be distinguished in terms of pairwise and multiple alignment, as reported in Table 1 and Table 2.

Another important aspect in which classical network alignment and network embedding alignment algorithms differ is related to the evaluation of the alignment quality, which is a basic task for the comparison of the different outputs. In fact, for both global and local and for both pairwise and multiple classical network alignment algorithms, new unifying alignment quality measures are developed to enable the fair comparison of different output types. On the other hand, there is not a unique framework for the evaluation of network embedding alignment algorithms. For example, REGAL is evaluated by using Alignment Accuracy, which considers the correct alignments among the total alignments. GeoAlign is evaluated by using the ICS, SPE, MNE, and COI measures; the alignment generated by MUNK is evaluated by applying the Gene Ontology consistency (GOC) measure and k-functional similarity. Protein2Vec is evaluated by using the ME, MNE, EC, ICS, and MNS measures and CONE-ALIGN is evaluated by applying the Alignment Accuracy. As is evident, the evaluation measures are different for each network embedding alignment algorithm. For this reason, efforts should focus on developing a framework, as in the case of classic alignment algorithms, for a unified comparison of the outputs between the different algorithms.

### Qualitative Comparison of NE-NA

A first rough classification of network embedding approaches in network alignment (NE-NA) can be made by differentiating the methods designed specifically for the alignment of PPI networks from the class of “domain-agnostic” algorithms also tested on PPI network datasets. In the first case, in addition to the topological information, the representation is enriched by considering biological similarities, while in the second case, only the topological information is sufficient to extract a representation.

Regardless of this first subdivision, each of the algorithms was made up of two consecutive phases: in the first phase, a vector representation of the nodes is determined, while in the second one, a matching map between the set of representations of the two or more considered networks is sought.

Following the taxonomy introduced in [7], regarding the NE step, the five considered algorithms can be divided into diffusion-based algorithms and matrix factorization algorithms. The difference, compared to the NE algorithms that are not compatible with the NA task, is that the NE-NA algorithms support the joint representation of two or more networks in the same vector space, and even if the representations are constructed separately in spaces of the same size, they are rejoined through semi-supervised approaches that exploit a set of landmarks.

With regard to the second step, the five considered algorithms can be divided into geometric (Geo) approaches, which leverage a rigid transport of the points of the first network on the points of the second one, or search or optimization (Opti) approaches based on the similarity of the vector representation of the nodes.

It is worth noting that according to the literature, none of the NE-NA approaches that leverage deep learning NE techniques, such as generative adversarial models or variational autoencoders, have been designed or tested on PPI networks.

The different characteristics of the considered NE-NA methods are summarized in Table 2, while a general pipeline of the considered PPI NE-NA algorithms is shown in Figure 2.

## 6. Conclusions and Future Works

In order to transfer biological knowledge from well-studied species, e.g., yeast, to less well-studied species, such as humans, a comparative analysis of PPI networks, well known as PPI network alignment, is a powerful means of inferring the functions of human proteins based on the functions of their aligned counterparts. Although many different network alignment methods have been proposed in biological fields, there are some issues related to network representation that still need to be solved. Network embedding has been recently proposed as a new approach for solving these issues. The goal is to learn a low-dimensional vector representation for network nodes so that the given similarity function is preserved as much as possible.

The present paper provides a review of the current representation learning approaches for addressing the problem of network alignment. An overview of classical network alignment techniques, their traditional classification, and the metrics commonly used to evaluate the quality of the alignment is provided. The problem of NE-NA in the context of PPI networks has been addressed and the main differences between the classical NA and NE-NA approaches have been discussed. Furthermore, five algorithms that leverage different network embedding techniques for the network alignment of PPI networks have been reviewed. The main contribution of this survey consists of addressing the problem of the NE-NA in the biological context of the PPI network. Because network embedding can offer great opportunities for network analysis taking advantage of the potential of machine learning techniques, in the future, we will focus on machine learning methods based on NE-NA in a wider biological context.

## Figures and Tables

**Figure 1 entropy-24-00730-f001:**
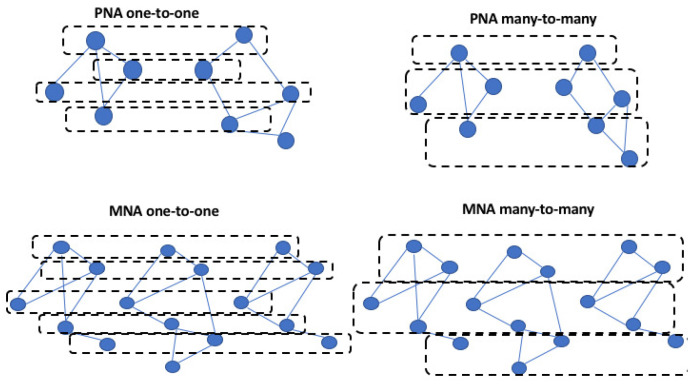
The figure shows an example of PNA one-to-one, PNA many-to-many, MNA one-to-one, and MNA many-to-many.

**Figure 2 entropy-24-00730-f002:**
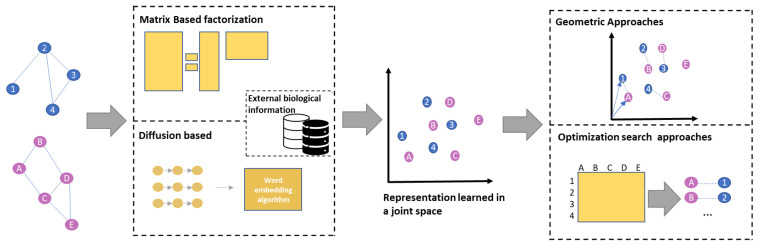
The figure shows the proposed taxonomy of the considered NE-NA algorithms for PPI networks based on their general pipeline.

**Table 1 entropy-24-00730-t001:** Network alignment algorithms.

Algorithm	GNA or LNA	PNA or MNA	One-to-One or Many-to-Many	Evaluation Measures
GRAAL	GNA	PNA	One-to-one	P-NC, R-NC, F-NC, GS3, NCV-S3, GO correctness, P-PF, R-PF, F-PF
H-GRAAL	GNA	PNA	One-to-one	P-NC, R-NC, F-NC, GS3, NCV-S3, GO correctness, P-PF, R-PF, F-PF
MI-GRAAL	GNA	PNA	One-to-one	P-NC, R-NC, F-NC, GS3, NCV-S3, GO correctness, P-PF, R-PF, F-PF
C-GRAAL	GNA	PNA	One-to-one	P-NC, R-NC, F-NC, GS3, NCV-S3, GO correctness, P-PF, R-PF, F-PF
L-GRAAL	GNA	PNA	One-to-one	P-NC, R-NC, F-NC, GS3, NCV-S3, GO correctness, P-PF, R-PF, F-PF
IsoRank	GNA	PNA	One-to-one	P-NC, R-NC, F-NC, GS3, NCV-S3, GO correctness, P-PF, R-PF, F-PF
GHOST	GNA	PNA	One-to-one	P-NC, R-NC, F-NC, GS3, NCV-S3, GO correctness, P-PF, R-PF, F-PF
WAVE	GNA	PNA	One-to-one	P-NC, R-NC, F-NC, GS3, NCV-S3, GO correctness, P-PF, R-PF, F-PF
MAGNA	GNA	PNA	One-to-one	P-NC, R-NC, F-NC, GS3, NCV-S3, GO correctness, P-PF, R-PF, F-PF
MAGNA++	GNA	PNA	One-to-one	P-NC, R-NC, F-NC, GS3, NCV-S3, GO correctness, P-PF, R-PF, F-PF
SANA	GNA	PNA	One-to-one	P-NC, R-NC, F-NC, GS3, NCV-S3, GO correctness, P-PF, R-PF, F-PF
IGLOO	GNA	PNA	One-to-one	P-NC, R-NC, F-NC, GS3, NCV-S3, GO correctness, P-PF, R-PF, F-PF
NetworkBLAST	LNA	PNA	Many-to-many	P-NC, R-NC, F-NC, GS3, NCV-S3, GO correctness, P-PF, R-PF, F-PF
NetAligner	LNA	PNA	Many-to-many	P-NC, R-NC, F-NC, GS3, NCV-S3, GO correctness, P-PF, R-PF, F-PF
AlignNemo	LNA	PNA	Many-to-many	P-NC, R-NC, F-NC, GS3, NCV-S3, GO correctness, P-PF, R-PF, F-PF
AlignMCL	LNA	PNA	Many-to-many	P-NC, R-NC, F-NC, GS3, NCV-S3, GO correctness, P-PF, R-PF, F-PF
LocalAli	LNA	PNA	Many-to-many	P-NC, R-NC, F-NC, GS3, NCV-S3, GO correctness, P-PF, R-PF, F-PF
GLAlign	LNA	PNA	Many-to-many	P-NC, R-NC, F-NC, GS3, NCV-S3, GO correctness, P-PF, R-PF, F-PF
MultiMAGNA++	GNA	MNA	One-to-one	NCV-MNC, NCV-CIQ, LCCS, MNE, GC, P-PF, R-PF, F-PF
GEDEVO-M	GNA	MNA	One-to-one	NCV-MNC, NCV-CIQ, LCCS, MNE, GC, P-PF, R-PF, F-PF
IsoRankN	GNA	MNA	Many-to-many	NCV-MNC, NCV-CIQ, LCCS, MNE, GC, P-PF, R-PF, F-PF
SMETANA	GNA	MNA	Many-to-many	NCV-MNC, NCV-CIQ, LCCS, MNE, GC, P-PF, R-PF, F-PF
LocalAli	LNA	MNA	Many-to-many	NCV-MNC, NCV-CIQ, LCCS, MNE, GC, P-PF, R-PF, F-PF
NetCoffee	GNA	MNA	Many-to-many	NCV-MNC, NCV-CIQ, LCCS, MNE, GC, P-PF, R-PF, F-PF
BEAMS	GNA	MNA	Many-to-many	NCV-MNC, NCV-CIQ, LCCS, MNE, GC, P-PF, R-PF, F-PF

**Table 2 entropy-24-00730-t002:** Characteristics of different NE-NA algorithms. The abbreviations “Geo” and “Opti” stand for geometric and optimization alignment approaches, respectively.

Algorithm	PNA or MNA	Output	Embedding Approach	Alignment Approach	External Biological Sources	Comparison Metrics
REGAL	MNA	Matching node list	Matrix factorization	Geo	Domain-agnostic	AA, α-AA, Runtime
GeoAlign	PNA	Matching node list	Matrix Factorization	Geo	Network simplex Sequence similarity	ICS, SPE, MNE, COI
MUNK	PNA	MUNK similarity score matrix (nxm)	Matrix factorization	Opti	Sequence homologs GO annotations	GO Con, K-fs AUPR
Protein2Vec	MNA	Similarity ranking	Diffusion method	Opti	BLASTP	ME, MNE, EC, ICS, MNS
CONE-ALIGN	PNA	Alignment matrix	Matrix decomposition	Geo	Domain-agnostic	AA

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
