# Peer review of "An Extensive Assessment of Network Embedding in PPI Network Alignment"

_entropy, 2022, doi:10.3390/e24050730_

Round 1

Reviewer 1 Report

The manuscript is a review of methods for network alignment via embedding into a latent space. It starts with an overview of classical network alignment tools, too. I think this is a nicely written paper on an important topic, and my comments are primarily related to the presentation.

  1. Formatting into paragraphs is questionable on many occasions: I think many paragraphs are too short and should be merged together. Examples are lines 224-254, 273-300, 387-407, 418-425, 521-530.
  2. Additionally, in Section 2 in each of the subsections the methods should be grouped be sharing common traits, which should be discussed. Now it is quite confusing, and the reader does not get a good impression of the trends in the field (at least I did not).
  3. Definition of network alignment in lines 186-188 somehow misses the issue of common network topology completely and focuses only on node similarity. Please fix, this is very important.
  4. Lines 23-26: I disagree that human is a poorly studied organism.
  5. Lines 58-62 and further in the text: Please be careful about your capitalization, there is no need to capitalize every term that you are going to use an abbreviation for.
  6. Line 83: a space is missing.
  7. Lines 114-115: Two 'moreover' in two consecutive sentences.
  8. Line 120: 'illustrating' -> 'illustrate'
  9. Line 182: What is a 'system data level'?
  10. Line 202: What happens if the number of nodes in the two networks differ?
  11. Line 412: What are n and m?
  12. Equations between lines 415 and 416: I get confused here about whether D_12 and S are the same matrix or not?

Author Response

The manuscript is a review of methods for network alignment via embedding into a latent space. It starts with an overview of classical network alignment tools, too. I think this is a nicely written paper on an important topic, and my comments are primarily related to the presentation.

  1. Formatting into paragraphs is questionable on many occasions: I think many paragraphs are too short and should be merged together. Examples are lines 224-254, 273-300, 387-407, 418-425, 521-530.

Answer: We thanks the reviewer to pointing out this. We formatted the paragraphs.

  1. Additionally, in Section 2 in each of the subsections the methods should be grouped be sharing common traits, which should be discussed. Now it is quite confusing, and the reader does not get a good impression of the trends in the field (at least I did not).

Answer: We apologize since we were not able to clarify the Section 2 . We re-organized Section 2. We discussed the state-of-arts  according the works that introduced only the representation learning and the network representation learning and then according  the works that proposed a taxonomy  for machine learning on graphs and the analytics task. Furthermore, we presented the work according the year.

  1. Definition of network alignment in lines 186-188 somehow misses the issue of common network topology completely and focuses only on node similarity. Please fix, this is very important.

Answer: We thanks the reviewer to pointing out this.   We rewrote the sentence ->

Network alignment is a common
problem that requires to search how best fits one network into another network. This “fit” can be quantify by assigning the amount of conserved, hence, aligned edges. Otherwise, measure of node conservation (NC) quantify the similarity between pairs of nodes from different networks . That is, intuitively, a good alignment should both map similar nodes to each other and preserve many edges. Thus, the problem of graph alignment consists of the mapping between two or more graphs to maximize an associated cost function  (also known as quality of alignment) that represents the similarity among nodes or edges.
Conventionally, given two graphs, G1 = {V1, E1} and G2 = {V2, E2}, the graph alignment
searches an alignment function f : V1 → V2, where V1,2 are set of nodes, that maximizes
the quality of alignment. The size of this search space is large, as it consists of all possible
mappings between nodes of the compared networks. The computational intractability of
the above problem, which arises from the NP-completeness [15] of the underlying subgraph sub-graph isomorphism issue, requires development of heuristics (approximate approaches) to solve the problem.

  1. Lines 23-26: I disagree that human is a poorly studied organism.

Answer: We thanks the reviewer to pointing out this.   We rewrote the sentence -> This enables to transfer biological knowledge  from well-studied species, i.e  yeast, to species whose studies are still ongoing such as human.

  1. Lines 58-62 and further in the text: Please be careful about your capitalization, there is no need to capitalize every term that you are going to use an abbreviation for.

Answer: We thanks the reviewer to pointing out this.   We fixed this by removing the capitalization.

  1. Line 83: a space is missing.

Answer: We apologize for the issue. We fixed this.

  1. Lines 114-115: Two 'moreover' in two consecutive sentences.

Answer: We apologize for the issue. We fixed this.

  1. Line 120: 'illustrating' -> 'illustrate'

Answer: We apologize for the issue. We fixed this.

  1. Line 182: What is a 'system data level'?

Answer: We thanks the reviewer to pointing out this.  We rewrote the sentence ->

Network alignment (NA) is a computational technique  widely used for comparative analysis of PPI networks between species, in order to discover the similar parts between molecular systems of different organisms.

  1. Line 202: What happens if the number of nodes in the two networks differ?

Answer: We thanks the reviewer to pointing out this. In general, alignment allows you to align two networks with the same number of nodes or with different number of nodes as long as the network with a smaller number of nodes is aligned with the network with a greater number of nodes.

  1. Line 412: What are n and m?

Answer: We apologize since we were not able to clarify this point. N and m are the order  of the initial matrices built by MUNK. We added this detail in the paper.

  1. Equations between lines 415 and 416: I get confused here about whether D_12 and S are the same matrix or not?

          Answer: We apologize for the issue. We fixed this by replacing D_12 with S.

Reviewer 2 Report

TO AUTHORS

The article by Milano and Cannataro analyzes the PPI network alignment problem describing the characteristics and differences between the classical approaches and network embedding algorithms and how these methods manage the alignment. The authors put together data on controversial topics by citing in many cases adequately studies on the role of these different approaches. The main contribution of this article consist of addressing the problem in the biological context of the PPi network adequately.  I think that this article can be accepted for publication.

Best Regards

Author Response

We thanks the reviewer for the positve comment.